# Traumatic Stress of Frontline Workers in Culling Livestock Animals in South Korea

**DOI:** 10.3390/ani10101920

**Published:** 2020-10-19

**Authors:** Hyomin Park, Myung Sun Chun, Yunjeong Joo

**Affiliations:** 1Department of Urban Sociology, University of Seoul, 163 Seoulsiripdaero, Dongdaemun-gu, Seoul 02504, Korea; hyominp@uos.ac.kr; 2Research Institute for Veterinary Science, College of Veterinary Medicine, Seoul National University, 1 Gwanak-ro, Gwanak-gu, Seoul 08826, Korea; jdchun@snu.ac.kr; 3The Institute for Social Development and Policy, Seoul National University, 1 Gwanak-ro, Gwanak-gu, Seoul 08826, Korea

**Keywords:** culling, mental health, post-traumatic stress disorder, PTSD, depression, animal disease, PANAS, animal attitude scale, AAS, animal ethics

## Abstract

**Simple Summary:**

Though culling is an effective measure for controlling animal diseases, it could have detrimental effects on mental health among frontline workers, and poses ethical problems concerning the treatment of animals. This research investigates the stress from culling and its effect on workers’ mental health. The data from an online survey of frontline workers showed that the working condition was very hard on them, causing adverse effects such as depression and post-traumatic stress disorder (PTSD) symptoms. Three-fourths of the respondents were identified as being from a high-risk group in terms of mental health. Further analysis revealed sequential causation, where workers with greater pro-animal attitudes felt more hardship in performing culling jobs, which triggered stronger negative emotions and resulted in higher PTSD scores. A regulation to implement processes for reducing the impacts of culling and ex-post measures for the workers is required to prevent the negative effects of culling on workers.

**Abstract:**

The last decade brought several devastating outbreaks of foot and mouth disease and avian influenza in South Korea, which had been handled through preventive culling, despite the controversy surrounding its efficiency and ethical considerations. Notably, the lack of regulations on culling processes has exposed the workers to extremely harsh working conditions. This study investigates the effect of culling jobs on the mental health of the frontline workers, based on 200 samples collected through a web-based survey conducted on participants with experience of culling tasks. Culling was found to have a powerful negative effect on the workers’ mental health, including high depression rates. Of those surveyed, 83.7% answered that the working conditions were intense, and 74.5% showed scores above the cutoff point for post-traumatic stress disorder (PTSD). A regression analysis revealed that individual’s attitudes toward animals mediated the effect of culling experience on PTSD symptoms. However, mental health care for the workers has been insufficient: 70.2% of the respondents were willing to get mental treatment to deal with the distress they underwent from culling. We conclude that engagement in culling has a detrimental effect on the workers’ mental health, and that they should be provided with systematic mental health care.

## 1. Introduction 

Culling livestock is a primary interventive measure for stamping out epizootic diseases in the modern livestock industry. For instance, during South Korea’s biggest foot and mouth disease (FMD) outbreak in 2010–2011, about 3.3 million pigs, 150,800 cattle, and 10,000 goats, as well as other animals, were slaughtered for preventive purposes. In the case of avian influenza (AI), more than 71 million poultries were killed between 2011 and 2017. To control these outbreaks, about 488,000 central and local government officials, 330,000 soldiers, and 146,000 police officers were mobilized for culling, in addition to about 700,000 civilians who were called as workers.

Preventive mass killing of animals, mostly without the disease, is a dreadful experience for both animals and humans. Though culling livestock animals is viewed as an effective method from the viewpoint of economic and scientific priorities to stop the rapid spreading of epizootics [1], there were many criticisms of the process concerning animals’ suffering from slaughter during the FMD epidemic [2]. Another problem of mass culling is its detrimental effect on human health. The poor working conditions of culling created many victims. Officially, 237 persons were physically injured, and 11 among them died during the 2014–2015 FMD outbreak in South Korea [3]. Recently the livestock farmers and workers were overloaded with the culling of millions of fowl due to AI, which has become a seasonal epidemic in the Korean peninsula.

Culling requires many animals to be killed under an outbreak situation, in a manner that often breaches humane slaughter regulations. The restrictions on movement, food shortage, and the lack of proper veterinary care for livestock designated to be slaughtered causes severe harm to the animals [4]. For this reason, ethical and social issues have been raised about the mass culling policies of countries with culling experiences. Mepham criticized the practice that the utilitarian basis of the mass slaughter program is no longer justified because of its long-term harm for human and animal welfare in the United Kingdom (UK) [5]. A study on the views held about future strategies for animal disease policies among the stakeholders from 24 member states in the European Union found that priority was placed on preventive measures, such as providing imperative information to stakeholders and vaccinations, which could prevent human trauma and the culling of animals, as well as the social, psychological, and financial consequences [6]. In the Netherlands, public concerns were raised about the psychological and economic damages to the farmers caused by mass culling policies, expressing negative opinions about preventive culling during the epizootics [7]. In South Korea, public criticism and distrust of the routine stamping-out policy have been raised from the perspective of bioethics, that is, the ethical issues surrounding medical processes, biotechnology, and governance, etc. [8,9]. The issue of ethics could also affect individuals who engage in ethically controversial practices like culling on a personal level, in the form of moral distress, as can be seen from a previous study showing that veterinarians experienced moral distress from internal ethical conflicts involved in contemporary veterinary practice, which negatively impact daily practice and life [10].

It is, indeed, a fact that repeated epizootic outbreaks and the unavoidable mass slaughtering of animals have caused helplessness and suffering among those who are involved in the disease control and prevention process, including farmers and government officials. The 2001 FMD outbreak in the United Kingdom affected the mental health of farmers and those involved in tourism [11]. A study based on the diary entries written by community members who had experienced the FMD crisis exhibited feelings of distress, grief, and fear of recurrence [12]. High levels of depression were measured by the General Health Questionnaire in the farmers of the locales affected by FMD, even among those who had not experienced slaughter on their farms [13]. In the Netherlands, after the FMD crisis, when 27,000 farm animals were culled, one-half of the investigated farmers suffered from post-traumatic stress [14], and the psychological impact of the FMD crisis was most critical among the Dutch farmers in the culled areas [15].

In this study, we investigated the psychological influences of culling on the mental health of the people engaged in the process. As mentioned above, previous studies on culling reveal that the work is excessively stressful, and has various negative impacts on the mental health of the farmers and frontline workers. In relation to such negative impacts, we focus on examining the level of stress experienced by frontline culling workers using the post-traumatic stress disorder (PTSD) scale and depression, through a web-based survey among local government officials and veterinarians who have experiences in mass culling in South Korea.

## 2. Culling and Trauma

Previous research indicated the detrimental effect of culling on mental health among people engaged in the process. In particular, the frontline workers handling the epizootic outbreaks were revealed to work in a dangerous and highly stressful environment without proper training, practical experience, or systematic treatment to deal with the stress from the work. As a result, they suffered from emotional exhaustion, hopelessness, fatigue, anger, and burnout [16,17]. Existing research has also revealed that these workers felt morally compromised by having to mass-slaughter animals, experiencing intensive compassion fatigue from the job [18]. The trauma from engaging in culling continues for years after the experience. In Japan, after the mass culling of about 290,000 livestock animals in 2010, 37.6% of the 875 workers who had been engaged in the job suffered from mental stress, and 11.5% of the respondents (97/843) still experienced post-traumatic stress at the time of the survey in 2012 [19].

In South Korea, the distress felt by the government officers and workers from the culling process was partially investigated. Kim and Hyun found that the 406 governmental officials who had engaged in the culling and disposal of livestock animals experienced feelings of helplessness and horror as traumatic events [20]. Of these, 34.5% fell under a high-risk group for PTSD, and 16.2% showed above-mild levels of depression. In a survey of 167 males in FMD outbreak areas, the maladaptive emotion regulation strategies of participants in culling predicted a high risk of PTSD, and they experienced other mental problems, which evidenced the necessity of proper intervention [21].

Previous studies on the mass culling trauma have focused on the level of trauma and the intensity of the dreadful experience. However, direct and indirect experiences of mass culling animals could also ruin normal human–animal relations at farms. Even in intensive livestock production and consumption systems, human–animal ties are based on preference, care, and proximity [22]; thus, the negative interactions may cause psychological harm to both sides. During the culling process, farmers suffer from feelings of dehumanization and insensitivity [23], and slaughterhouse workers experience heightened emotive responses and personality changes [24]. Animal workers (such as veterinary teams, animal research staff, and shelter staff) who generally show affinity toward animals experience traumatic symptoms when killing animals [25]. Therefore, personal attitudes towards animals and previous relations with animals significantly impacted the frontline workers’ distress and trauma levels.

To measure the overall distress from culling in various aspects, we examine the general stress level and factors that moderate the relationship between the culling experience and PTSD symptoms among frontline workers in the culling process. This research intends to gain a comprehensive understanding of the general distress experienced by culling workers rather than focusing on a specific dimension. We also check the emotional valences felt by the workers during culling and their behavioral impact, in order to check the impact of mass culling on the workers’ stress levels.

## 3. Methods

### 3.1. Participants

To measure the stress level and other influences of the culling process, we performed an online survey among government officials and veterinarians who had participated in culling during the epizootic outbreaks. The questionnaire consisted of four main sections: experiences from the culling process, psychological effects, cognitive responses, and behavioral reactions. The survey was performed via a commercial Internet survey platform, and a total of 200 valid cases were collected.

All respondents were Koreans residing in South Korea, and the survey was performed in the Korean language. The respondents’ average age was 34.9 years old, and 89.0% were male. About half of the respondents (51.0%) were local government officials in charge of animal husbandry and health control, 25.0% were public veterinarians, 19.7% were supporting officials from other departments whose tasks are not related to animals, and 1.5% were public health officials. Regarding culling experiences, 15% of respondents participated in culling jobs only once, 29.5% participated two or three times, 11.5% participated three or four times, and 44.0% participated more than six times. In terms of the animals they had handled, 70% of the respondents had worked with chickens, 53% with cows, 37.5% and 29.5% with pigs and ducks, respectively, while 14.5% participated in culling other animals, such as dogs, cats, geese, goats, ostriches, and deer (multiple answers were allowed)(see, Figure 1).

### 3.2. Measurements

The questionnaire was structured in the following order. First, the respondents were asked about their experiences of culling and the working conditions during culling. Then, questions about their stress and depression levels, emotional reactions, attitudes towards animals, and evaluation of the government’s measures for animal disease were presented. Lastly, the demographics of respondents were specified. We used PTSD and other psychological scales to examine how the respondents’ involvement in culling affected their psychological states. To measure PTSD, we used the Impact of Event Scale-Revised in Korean (IES-R-K). This scale was originally developed by Weiss and Marmar [26], and was translated and standardized in Korean [27]. The scale consists of 22 items to measure three main symptoms of PTSD: intrusion (eight items), avoidance (eight items), and hyperarousal (six items). Each item ranges from 0 to 4 points, and the suggested cutoff point of the scale is 24/25 points. The reliability of the scales is reported as 0.971.

Depression is one of the main symptoms that accompanies PTSD. In this study, we used the Beck Depression Inventory (BDI) [28], a commonly-used depression scale for self-report surveys. This study used the third version of BDI to check the respondents’ levels of depression. This version now includes measures for the feelings of agitation and worthlessness, concentration difficulty, and the loss of energy, following the Diagnostic and Statistical Manual of Mental Disorders [29]. The scale consists of 21 items, each item ranging from 0 to 4 points, and the suggested cutoff point of the scale is 18 points. The reliability of the Korean-translated scale is 0.94 [30].

Besides the levels of PTSD and depression, we tried to grasp the culling process’s affective influence, assuming that positive and negative affective mood states also indicate the event’s impact. For this, we used the Positive and Negative Affect Schedule (PANAS), which is a self-reported questionnaire for positive and negative emotions. Each valence has ten items, which are rated on a scale of 1 to 5 to indicate the extent to which the event influenced the respondents. The original scale was introduced by Watson et al. [31], and the version used in this study was translated into Korean by Lee et al. [32], and recently revised by Park and Lee [33]. The scale’s reliability is reported as 0.83 (positive affect = 0.86, negative affect = 0.83). The respondents were asked to recall emotion that they felt during the culling when answering the questions.

We assumed that the respondents’ attitudes toward animals would regulate the intensity of the psychological influences of culling. We used the brief measure of the Animal Attitude Scale (AAS-5) developed by Herzog et al. [34] to measure the respondents’ attitudes. Though the scale was reduced from the original version of 20 items to 5 items to lessen the burden on the respondents, it shows excellent properties and offers alternatives by demonstrating acceptable reliability (Cronbach’s alpha > 0.80) and a strong correlation with the original version (*r* > 0.95, *p* < 0.001) [35].

Other than the psychological scales, the survey also checked the respondents’ demographic characteristics: gender, age, education, occupation, etc. The respondents were also asked to specify their experiences in culling processes, such as the intensiveness of the tasks they had handled, the number of times they had participating in culling, and the species of the animals they had been involved in culling.

## 4. Results

### 4.1. Working Conditions

To examine the intensity of culling processes, we asked how hard the working conditions were using an 11-point scale (1 for “very easy”, 6 for “hard”, and 11 for “very hard”). The average score for working conditions was 9.02, indicating that many people thought their tasks were hard or very hard. Only 16.3% answered lower than the middle point of the scale (6 points), while 83.7% answered higher than the middle point, showing that their overall working conditions were very intense.

### 4.2. Affective States

In addition to evaluating working conditions, we examined the respondents’ emotional reactions in the culling process. We used the PANAS scale to test the level of positive and negative affect from culling tasks. The scores of our respondents were compared with the scores from a previous study validating the scale [36], which was based on the data of 300 ordinary respondents in South Korea as a reference point to statistically test the differences between the respondents of the two studies. We performed a *t*-test using STATA 16 program.

The results showed that the scores for almost all positive affective states, such as excitement, pride, interest, and activeness, were significantly lower than the reference scores (Figure 2a). In contrast, the intensity of negative affect, such as guilt, distress, irritation, and nervousness, was significantly higher than the reference scores (Figure 2b). Only “attentive” was not significantly different from the reference point. The results suggest that the culling experiences are not only stressful, but also harm the workers’ emotional states.

### 4.3. Depression

Depression is a chronic mood disorder reported as one of the main consequences of stressful events, and the comorbidity between depression and PTSD is very high [37]. BDI-II was used to test depression among the workers. The suggested cutoff points of the BDI-II for mild depression, moderate depression, and severe depression are the ranges of 14–19, 20–28, and 29–63, respectively [28]. The average score of the respondents in our survey was 14.53, which indicates borderline mild depression. Of the respondents, 15.0% exhibited scores between 20–28 (moderate levels of depression), and 17.0% showed severe depression (Table 1).

### 4.4. Trauma and PTSD

IES-R-K scores were analyzed to check the levels of PTSD among the respondents. The average score gained by the respondents was 41.31. Considering that the suggested cutoff point of the scale has been 25 points in previous studies, this score is significantly higher (*p* < 0.001). Only 25.5% of the workers received lower scores than the cutoff point, while the remaining 74.5% recorded scores above the cutoff point, meaning that three-fourths of the workers are at risk of PTSD and need further medical examination or treatment (Figure 3). This suggests that the tasks during the culling process were intense and stressful enough to induce PTSD among most workers.

The scores for the IES-R-K vary depending on individual characteristics. By gender, female respondents recorded 45.9 points on average, which is significantly higher than the male respondents’ scores (40.7). However, the difference was marginally insignificant (*t* = 1.179), which was mainly attributed to the small number of female samples. The attitude toward animals was also found to affect the score. The association between respondents’ attitudes toward animals and their level of PTSD was measured using a simplified Animal Attitude Scale (AAS-5) [34], and the results show that the attitude toward animals was positively correlated with PTSD scores (*r* = 0.219, *p* < 0.002). That is, people with a more positive attitude toward animals experienced higher levels of PTSD symptoms.

### 4.5. Regression Analysis for PTSD

We performed a regression analysis to examine the factors that affect PTSD symptoms. The models included evaluations of working conditions, attitudes toward animals, depression, affective states, and other demographic variables. According to our analysis, working conditions and attitudes toward animals were identified as antecedents predicting the levels of PTSD, and depression was also a significant factor affecting PTSD scores. However, other demographic variables, such as age and gender, did not have an effect on PTSD. It is also noteworthy that the number of participations in culling had an insignificant effect on PTSD (see Table 2).

Notably, in the second model, we examined if affective states mediated the relationship between attitude toward animals and PTSD. The results showed that the level of negative affect influences the level of PTSD. When affective state was included in the model, the effect of AAS was reduced to an insignificant level, indicating its full mediation between the attitude toward animals and PTSD. That is, if he/she had a higher score in the Animal Attitude Scale (AAS), they are more likely to feel negative emotions, and thereby, were more likely to suffer from PTSD symptoms. In addition, we found that the emotions felt during the culling process partially mediated the effect of working conditions on the PTSD score in model 2.

Interestingly, affective states showed an asymmetrical impact on PTSD. In accordance with previous research on the asymmetry between positive information and negative information [32], our analysis showed that the evocation of negative emotions is strongly associated with PTSD, while reducing positive emotions had a weak relationship in increasing PTSD.

## 5. Discussion

This study found that frontline culling workers underwent feelings of guilt, distress, and irritation very intensively. This study also confirmed the detrimental effects of engaging in culling on workers’ mental health. Most of the workers thought the working condition was challenging, and many of them felt depression, which in turn made them feel negative emotions during the process. Almost three-fourths of the respondents reported they were suffering from light or severe symptoms of PTSD. Though we did not check their PTSD scores before culling, this percentage is quite extraordinary considering that the lifetime prevalence rate of PTSD is reported to be 8.7% in the United States [29] and 1.6% in South Korea [37]. Thus, it is important to understand the various components of negative emotions related to culling and PTSD to determine effective measures for reducing negative affect. The most effective way to prevent workers from experiencing PTSD from culling would be to stop using this method for livestock control. However, the occurrence of widespread livestock diseases makes it inevitable for culling to be conducted on a mass scale to prevent further damage. Culling is widely accepted as a necessary measure for preventing the spread of animal diseases [38]. In such an inevitable situation, it is important to reduce the distress and pain caused by culling on both humans and animals. To do this there should be a systematized pre-operational education and training about the purpose and process of culling for potential frontline workers, who need to understand the inevitability of the hard and stressful tasks to protect uninfected animals and the safety of the community.

This research also found that the number of participations in culling did not affect PTSD, which indicates the importance of implementing interventions in every instance for the amelioration of PTSD. Age and gender also did not influence PTSD much; females are more vulnerable, to a statistically insignificant degree, possibly due to the small female sample (11%).

According to the regression analysis for PTSD, working conditions, AAS, and depression played important roles in deciding the PTSD score for each individual. This implies the need to curtail negative emotions produced by negative culling experiences, such as witnessing the painful death and violent handling of animals and harsh working conditions.

It is also noteworthy that the effect of AAS on PTSD weakened when emotions were included in the model, suggesting that AAS score is strongly related to negative emotions among workers; people with high AAS are emotionally more vulnerable. The practical implication of this finding is that reducing negative affect during culling is more important in reducing the workers’ psychological impact from the job than enhancing positive affect.

## 6. Conclusions

The results of this research have many practical implications. As a supplementary question, in the survey, we asked if the workers wanted to receive mental treatment or participate in a counseling program concerning the stress they felt from culling processes; 70.2% of respondents answered yes, showing how intense their stress levels were and indicating the necessity of an aid program for the workers. Based on the results, we recommend that the government establish measures for these human victims of the culling process, especially in ways that consider the workers’ various situations and positions. More importantly, the government should prepare regulations geared toward reducing the workers’ stress from engaging in culling jobs on site. For instance, we found that the insufficient covering and treatment of animal carcasses required the workers had to handle them manually, causing huge stress to the workers. Moreover, in addition to governmental action, social attention should be paid to these workers, as there have been incidents of suicide due to the unbearable stress from culling. Such casualties could be reduced by building a social safety net for these culling workers.

This study attempted to visualize the damages incurred on the workers engaged in culling processes. One limitation of this research is its survey sample, which was mostly composed of public officials and workers who organize and control the culling site. We were unable to reach out to the manual workers who directly conducted culling and handled livestock carcasses, due to certain limitations; for instance, many of them are migrant foreign workers with fluid residences. However, we presume that migrant workers’ working conditions would be similar or even worse, since they are required to stay at the culling site for two or three days, isolated from the outside world. Incorporating the working conditions for foreign workers who have engaged in culling will enable a fuller assessment of culling’s psychological impact on frontline culling workers’ mental health. Another important dimension future studies should consider is moral distress. Veterinarians who are given the job of killing animals on mass scales experience moral distress, as such jobs come into conflict with their motivation as a vet to save the lives of animals. This dimension of moral distress should be included in further research in relation to the AAS. Furthermore, future research may also benefit from examining whether decreasing the pain felt by animals during culling is relevant to preventing negative emotions and PTSD, as well as the impacts of therapeutic experiences on negative affects and PTSD.

## Figures and Tables

**Figure 1 animals-10-01920-f001:**
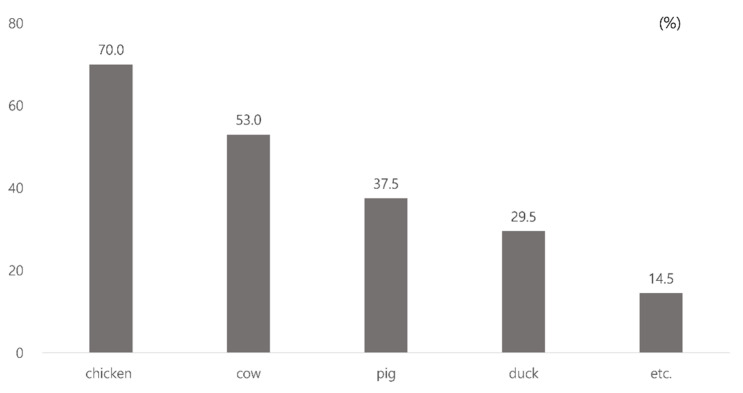
The types of animals handled in culling.

**Figure 2 animals-10-01920-f002:**
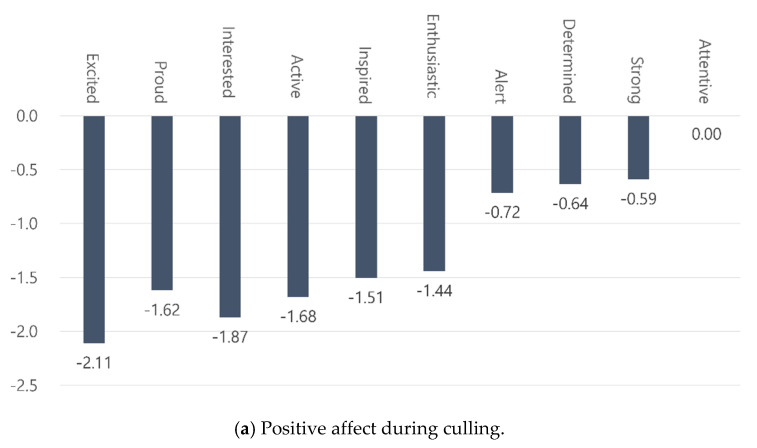
Differences in affective states compared to reference points: (**a**) decrease of positive affects felt during the culling process is much larger than (**b**) the increase of negative affects felt from the work.

**Figure 3 animals-10-01920-f003:**
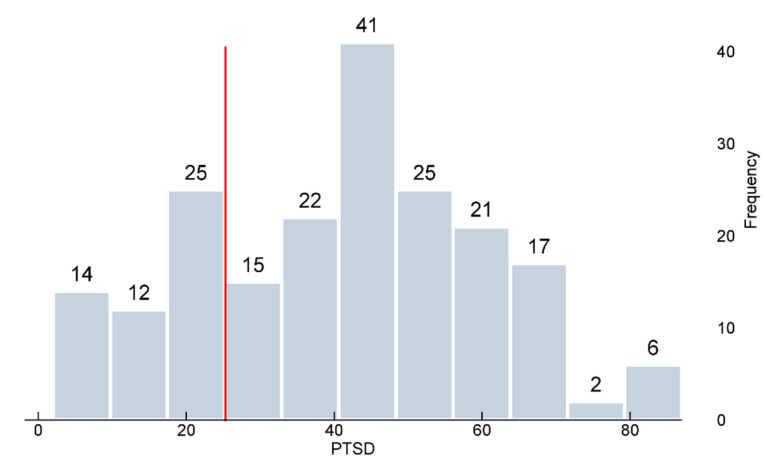
PTSD scores of frontline workers: 74.5% of the respondents received higher scores than the cutoff point (the red line).

**Table 1 animals-10-01920-t001:** Beck Depression Inventory (BDI)-II for depression.

Score	Diagnosis	No. of Respondents	%
0~13	Light Depression	117	58.5
14~19	Mild Depression	19	9.5
20~28	Moderate Depression	30	15.0
29~63	Severe Depression	34	17.0
Total		200	100.0

**Table 2 animals-10-01920-t002:** Results of OLS regression analysis on post-traumatic stress disorder (PTSD) scores.

	Model 1			Model 2		
PTSD	Coef.	Std. Err.		Coef.	Std. Err.	
Age	0.061	(0.097)		0.088	(0.091)	
Gender	2.534	(2.894)		1.562	(2.669)	
Number of participations	−0.016	(0.803)		0.450	(0.744)	
Working conditions	−7.859	(1.883)	***	−4.321	(1.827)	*
Animal Attitude Scale	3.971	(1.689)	*	1.010	(1.627)	
Depression	0.815	(0.072)	***	0.735	(0.070)	***
Negative emotion				6.052	(1.262)	***
Positive emotion				−2.904	(1.488)	
Constant	24.894	(9.712)	*	13.035	(10.367)	
*n*	200			200		
Adj. *R*-squared	0.566			0.633		
Model fitness	F_(6,193)_ = 44.17		***	F_(8,191)_ = 43.86		***

* *p* < 0.05, *** *p* < 0.001.

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
