# Peer review of "Traumatic Stress of Frontline Workers in Culling Livestock Animals in South Korea"

_animals, 2020, doi:10.3390/ani10101920_

Round 1
Reviewer 1 Report
The contribution addresses a very important topic and the study is interesting and useful. Nevertheless, it could be improved in the future including an analysis of the specific moral distress occurring during massive culling of animals.
Here are some general considerations:
- Moral distress is one of the most important factors to be considered when determining stress conditions like those described in the contribution. It is suggested to make reference at least in Introduction to moral distress, compassion fatigue and burn out, giving some hints about the choice not to include them in data collection.
- It is suggested to expand the discussion session
- It is suggested to include, in Conclusion paragraph, some sentences about the opportunity for developing a future investigation of the specific moral distress which is caused by massive culling, using appropriate scale
Below some detailed suggestions:
- I suggest to add ‘ethics’ or ‘ethical issues’ among keywords
- Line 74: please specify what it is meant here by ‘bioethics’
- Lines 59 ss: it is suggested to make reference to ‘moral distress’ as a specific human disease which is generally occurring in these situations and to add some references on the matter to be included in Bibliography
- Lines 98-99: same
- Line 99: what specific type of ‘fatigue’? Compassion fatigue is to be mentioned here
- Line 99: it is suggested to explain what ‘morally compromised’ in this scenario means; this is the key point to make reference to moral distress;
- Lines 99 and 102: I would mention also ‘burn out’ and add some references on that, if appropriate
- Lines 114 ss: I would add also reference to ‘care’
- Line 121: again it is important to consider also the moral distress occurred: it has to be explained more in detail why it has been chosen to ‘examine the general stress level’ and not the specific moral distress level, in order to properly justify the identification of the four main sections of the questionnaire
- Lines 161: why not compassion fatigue?
- Lines 198ss: it seems more appropriate to include this suggestion into discussion section
- Lines 226ss: same
- Lines 260ss: same
- Line 267: I suggest to open the discussion session with remarks made in lines 198ss, etc, expanding those considerations before to go to ‘This study also confirmed…’
Reviewer 2 Report
This is a well-written paper on an important research topic. The literature on the mental health of those working in animal-related fields is sparse. Frontline workers doing culling work are at risk for stress and trauma. This paper describes a study addressing this question. I appreciate the opportunity to review the manuscript and my comments are below:
Participants – the researchers should share the race or ethnicity of participants
Materials – I am more familiar with the term cuttoff point instead of cut point or cutting point of the scales.
Materials – PANAS instructions – please include the instructions for this scale. Did they report on their current affect or affect while they were doing the culling? Need clarification.
Materials – Why was the Animal Attitude Scale shortened to just 5 items? What is the evidence it is still a reliable and valid measure? Can you share analysis from this study showing any psychometric properties?
Materials – What was the order in which the scales were administered? Could be an order effect, if participants answer questions about culling and then complete scales on affect and mood.
Results – For the PANAS you state the scores were significantly different from the reference point but don’t name the statistical test or provide the actual results.
Results – Do you know anything about the participants’ mental health prior to the culling? There are substantial depression and PTSD rates in your sample but we don’t know whether they were ill already before culling or if symptoms occurred after culling. This is the most important critique. You’re suggesting you found evidence that culling can result in depression and PTSD, but it could be that they were ill beforehand. In your literature review, you suggest people who do this work may be at risk for mental health issues. Can you verify the participants did not meet diagnostic criteria before the culling?
Line 259 – you state that negative emotions had a strong effect on increasing PTSD. I suggest rephrasing this to say there was an association between the variables, or one variable was a significant predictor of the other. Using the term “effect of” suggests cause-and-effect. Your research design does not allow for that.
In addition, the paper should be read for minor errors in writing, such as:
Line – 98 - Systematic
Line 189 - PANAS
Line 191 - ..based [on] the data
Line 242 - We performed
Line 267 – this is the first line of the Discussion section. Using the word “also” seems out of place.
Line 310 - There seems to be a type-o in the sentence “The incorporate the working…”
